# The Protective Effects of the Autophagic and Lysosomal Machinery in Vascular and Valvular Calcification: A Systematic Review

**DOI:** 10.3390/ijms21238933

**Published:** 2020-11-25

**Authors:** Cédric H. G. Neutel, Jhana O. Hendrickx, Wim Martinet, Guido R. Y. De Meyer, Pieter-Jan Guns

**Affiliations:** Laboratory of Physiopharmacology, University of Antwerp, 2610 Antwerp, Belgium; cedric.neutel@uantwerpen.be (C.H.G.N.); jhana.hendrickx@uantwerpen.be (J.O.H.); wim.martinet@uantwerpen.be (W.M.); guido.demeyer@uantwerpen.be (G.R.Y.D.M.)

**Keywords:** vascular calcification, aortic valve calcification, autophagy, lysosomes, vascular smooth muscle cell (VSMC), valvular interstitial cell (VIC)

## Abstract

Background: Autophagy is a highly conserved catabolic homeostatic process, crucial for cell survival. It has been shown that autophagy can modulate different cardiovascular pathologies, including vascular calcification (VCN). Objective: To assess how modulation of autophagy, either through induction or inhibition, affects vascular and valvular calcification and to determine the therapeutic applicability of inducing autophagy. Data sources: A systematic review of English language articles using MEDLINE/PubMed, Web of Science (WoS) and the Cochrane library. The search terms included autophagy, autolysosome, mitophagy, endoplasmic reticulum (ER)-phagy, lysosomal, calcification and calcinosis. Study characteristics: Thirty-seven articles were selected based on pre-defined eligibility criteria. Thirty-three studies (89%) studied vascular smooth muscle cell (VSMC) calcification of which 27 (82%) studies investigated autophagy and six (18%) studies lysosomal function in VCN. Four studies (11%) studied aortic valve calcification (AVCN). Thirty-four studies were published in the time period 2015–2020 (92%). Conclusion: There is compelling evidence that both autophagy and lysosomal function are critical regulators of VCN, which opens new perspectives for treatment strategies. However, there are still challenges to overcome, such as the development of more selective pharmacological agents and standardization of methods to measure autophagic flux.

## 1. Introduction

Autophagy (“self-eating”) is a highly conserved catabolic process, responsible for the degradation of long-lived cytosolic proteins and organelles [1]. Three distinct forms of autophagy have been reported in mammalian cells: Macroautophagy, microautophagy and chaperone-mediated autophagy (CMA). While all three forms result in the delivery of cargo to the lysosomes for degradation, macroautophagy (hereafter referred to as autophagy) is the most investigated form [2,3]. The autophagic machinery is finely regulated, starting with generating an isolation membrane, known as the phagophore (Figure 1). The phagophore expands and engulfs intra-cellular cargo, resulting in the formation of a double-membraned structure, called the autophagosome. These autophagosomes fuse with lysosomes, forming autolysosomes that eventually degrade the autophagosomal cargo [3]. Consequentially, lysosomal function is closely entangled with autophagy function, as defective lysosomal activity impairs autophagic flux [4]. Additionally, autophagy can be specific in selecting its cargo for degradation. Pexophagy (peroxisomes), mitophagy (mitochondria), lipophagy (liposomes), ribophagy (ribosomes), xenophagy (intracellular pathogens) and aggrephagy (aggregated proteins) are examples of selective autophagy, mediated by specific autophagy receptors [5]. 

Multiple autophagy-related genes (*ATG*) have been identified, whose products are specific to the autophagic process [6]. Autophagy is regulated by these Atg proteins and mediated by upstream signalling, with the mechanistic target of rapamycin (mTOR) and AMP-activated kinase (AMPK) as the most important regulators. In short, mTOR represses autophagy induction by inhibiting unc-51 like autophagy activating kinase 1/2 (ULK1/2) and the vacuolar protein sorting 34 (VPS34) complex, while also preventing global expression of lysosomal and autophagy genes via phosphorylating transcription factor EB (TFEB) [7]. In contrast, AMPK, a cellular energy sensor, induces autophagy in low cellular energy conditions by both inactivating mTORC1 and phosphorylating ULK1 [8]. Additionally, AMPK induces autophagic flux by modulating both autophagosome maturation and autolysosome fusion [9]. The formation of autophagosomes relies on two ubiquitin-like conjugation systems: (1) microtubule-associated protein light chain 3 (LC3)-phosphatidylethanolamine (PE) (LC3 is the mammalian homolog of yeast Atg8) and (2) Atg5-Atg12 that have an essential role in autophagy and are widely conserved among eukaryotes [6,10]. In the first system, proLC3 is cleaved by the cysteine protease Atg4, forming LC3-I [11]. LC3-I is then conjugated with PE by Atg7 and Atg3 which are E1- and E2-homologous enzymes respectively, forming LC3-II that associates with autophagic membranes and is often used as a marker for autophagy [10,11]. In the second system, Atg5 is linked to Atg12 by the E1 and E2 enzymes Atg7 and Atg10, which further establishes a complex with Atg16L1 [11,12]. These Atg5-Atg12-Atg16L1 complexes oligomerize, forming larger units, which are required for targeting LC3 to the autophagic membranes [10]. These two conjugation systems are interdependent since LC3 conjugation is dependent on the activity of the Atg5-Atg12-Atg16L1 complex [10].

Autophagy has a central role in maintaining cellular homeostasis by eliminating unwanted, harmful cytosolic material [12]. The autophagic elimination of dysfunctional mitochondria or protein aggregates is necessary for mitigating the release of pro-apoptotic mediators and reactive oxygen species, safeguarding cell survival [11,13]. Accordingly, autophagy modulates many pathologies, playing an important role in cardiovascular diseases such as atherosclerosis, heart failure, arrhythmia, chemotherapy-induced cardiotoxicity, as well as being implicated in cardiac and vascular aging [14,15,16]. More recently, autophagy has been found to protect against vascular medial calcification, which is considered as an active, pathological and multifactorial phenomenon that is distinct from atherosclerosis [17,18,19,20,21,22]. Cardiovascular calcification features the deposition of calcium phosphate minerals, mainly hydroxyapatite [Ca_10_(PO_4_)6(OH)_2_] crystals, in cardiovascular tissues including arteries, heart valves, and the cardiac muscle [23,24,25]. Recent research showed that vascular smooth muscle cells (VSMCs) in the media can differentiate to osteoblast-like cells and form matrix depositions, making the process of vascular calcification (VCN) very similar to bone ossification [21,25,26,27,28]. VSMCs undergo osteogenic differentiation driven by increased intracellular concentrations of calcium and/or inorganic phosphate (Pi) [23,26,29]. An indispensable transcriptional regulator for this osteoblastic differentiation is the core-binding factor subunit 1α/runt related transcription factor, Cfba1/Runx2, causing VSMCs to release small phospholipid-bound matrix vesicles (MVs) [22,23,29,30,31]. Released MVs contain alkaline phosphatase and annexins that promote the deposition of hydroxyapatite crystals. Moreover, the Cfba-1 transcription regulator controls the secretion of bone-associated proteins, such as osteopontin, collagen type 1, osteoprotegerin and osteocalcin [23,26,32,33,34]. Besides numerous transcriptional and pro-osteogenic factors, bone morphogenetic proteins (BMPs) are upregulated within calcifying VSMCs [28,35]. The latter protein family consists of 15 members and forms a sub-group of the TGF-β super-family. These proteins play an important role in VCN and bone proliferation, development and fracture healing [36,37,38]. Within the BMP protein family, BMP2, BMP7 and BMP9 have been reported to have the greatest osteogenic-promoting activity [35,39,40,41,42]. BMP2 increases the uptake of Pi, promotes RANKL osteoblastic activity in VSMCs and induces VCN via the Wnt/βcatenin pathway [38,43,44,45,46]. 

Today, therapeutic options for treating VCN are limited. Targeting the autophagic and/or lysosomal machinery has been proposed as a promising strategy for drug development in VCN. This review aims to gather the current knowledge regarding the benefits and drawbacks of modulating autophagic-lysosomal function in VCN and aortic valve calcification (AVCN). Therefore, studies with pharmacological agents that modulate the autophagic-lysosomal machinery in VCN are listed. A recent review summarized the underlying mechanisms by which autophagy protects from VCN [47]. They broadly discussed how the protective effects of autophagy on VCN are linked with apoptosis, osteogenic differentiation and the release of MVs. However, they did not provide a systematic overview of how modulating autophagy, with existing pharmacological agents, affects VCN. 

## 2. Objectives

Lately, autophagy has gained a lot of interest in the VCN research field, with emerging evidence suggesting that autophagy plays a protective role by inhibiting apoptosis and MV secretion [48,49]. Different research groups are exploring autophagy as a treatment against VCN. Therefore, new and exciting data become available, that improve our understanding of autophagy and its possible pharmacological modulation in VCN. This review aims to provide a comprehensive overview of the current knowledge regarding the modulation of the autophagy-lysosomal pathway in both VCN and AVCN. Different interventions that either promote or inhibit the progression of VCN are compared and the underlying molecular targets are discussed. Additionally, this review will critically discuss the applied methods for evaluating autophagic activity.

## 3. Methods

### 3.1. Search Strategy 

The PRISMA guidelines were followed to assess the literature regarding the role of autophagy and lysosomal function in VCN [50]. The databases MEDLINE (PubMed Database), the Cochrane Library and the Web of Science were consulted up to 29 June 2020. The search string for the databases was: [{(Autophag*) OR (Autolysosome) OR (Mitophagy) OR (ER Phagy) OR (lysosomal)} AND {(Calcif*) OR (Calcinos*)} NOT (Atheroscler*)].

### 3.2. Inclusion and Exclusion Criteria

Studies were included if they investigated mechanisms of autophagy and/or lysosomal function in VCN, either through genetic alterations or (non-)pharmacological intervention. Both in vivo and ex vivo/in vitro animal and human studies were included. In vitro studies using cells other than vascular and valvular cells (VSMCs, valvular interstitial cells (VICs)), were excluded, as well as studies investigating other forms of pathological mineralization. The publication date was not an exclusion criterion. The reference lists of the included articles were checked to identify additional relevant studies. Only primary research articles were used and therefore abstracts and reviews were excluded. Furthermore, only English language articles were included.

### 3.3. Data Extraction

For the included articles, the type of modulation, the methods, the molecular findings and the study conclusion were extracted. The types of modulation regarding the autophagy-lysosomal pathway are divided in three categories, namely: (1) (non)-pharmacological modulations, (2) genetic modulations, and (3) no intervention/modulation (=observational).

## 4. Results

### 4.1. Study Selection

The search strategy resulted in a combined number of 602 articles, extracted from all three databases (Figure 2). After removing all duplicates, 457 articles were manually checked for relevance by screening their titles and abstracts. Forty-six articles were selected, of which 10 were excluded after full text review. Through the review of the reference lists of the included articles, an additional reference was found eligible and was included. Hence, 37 articles were included for this systematic review.

### 4.2. Study Characteristics

The characteristics of the included studies are presented in Table 1 and Table 2. Most of the studies were published in the time period 2015–2020 (92%), with 58% of the studies published between 2019 and 2020. Regarding the studied matrix, 32 studies (89%) studied aortic/VSMC calcification while four studies (11%) studied valvular calcification. Among the studies investigating aortic/VSMC calcification, 26 (82%) studies investigated the autophagic machinery in calcification, while six (18%) studies explored lysosomal function in VCN. In total, seventeen studies conducted both in vitro and in vivo research, while sixteen studies solely conducted in vitro research and two studies solely in vivo research. Three studies conducted ex vivo research with tissue from human patients. Overall, most studies used different pharmacological and/or genetic approaches to study the effects of modulating autophagy or lysosomal function in VCN. Therefore, together with the significant heterogeneity regarding methods and output analysis, it was not possible to perform a meta-analysis on these data.

### 4.3. Autophagy in Vascular Calcification

The studies included in the current review focused, either on modulating (induction or inhibition) autophagy in calcifying conditions (i.e., high Pi and/or high calcium), or on describing how a pro-calcific environment affects autophagic flux or a combination of both (Table 1). Induction of autophagy was established through mTOR inhibition, AMPK activation or unspecified mechanisms stimulating autophagic flux. Additionally, different studies used autophagy inhibitors, either by itself or in combination with an autophagy inducer to demonstrate the involvement of autophagy.

#### 4.3.1. The Effect of Vascular Calcification on Autophagic Activity

Autophagic flux was upregulated in VCN in both a chronic in vivo renal failure rat model, as well as in an in vitro Pi-induced VSMC calcification assay [51]. This observation was also confirmed in other studies, where autophagy was upregulated in in vitro and in vivo models of VCN [52,53]. Conversely, there is also evidence that mTOR (an inhibitor of autophagy) is upregulated in VSMC calcification [54]. In line with these findings, epigenetic regulation of mTOR activity was also altered in VCN. Histone deacetylase 1 (HDAC1) was downregulated in VCN while lysine-specific histone demethylase A1 (LSD1) was upregulated, resulting in enhanced mTOR activity [52]. Lastly, hydroxyapatite was shown to cause cell injury, leading to decreased lysosomal integrity and facilitating VCN [55].

#### 4.3.2. Stimulating Autophagy in Vascular Calcification through mTOR Inhibition

Inhibition of mTOR with rapamycin, adiponectin, bavachin or changing epigenetic regulation of mTOR (through HDAC1 and LSD1) consistently ameliorated VCN. Rapamycin and adiponectin, two known mTOR inhibitors, strongly attenuated VSMC calcification in vitro [54]. In line with these findings, mTOR overexpression aggravated VCN, whereas kinase-dead mTOR expression diminished VCN [56]. (Over)expression of mTOR supressed *Klotho* expression, whereas rapamycin rescued this suppression and alleviated VCN. Since *Klotho* knockout mice developed aggravated VCN, it was proposed that mTOR promotes VCN through suppression of *Klotho* [56]. Furthermore, the flavonoid bavachin ameliorated VCN by inhibiting mTOR, whereas wortmannin or Atg7 knockdown abrogated the protective effects of bavachin in vitro [57]. Lastly, HDAC1 was downregulated in VCN, while LSD1 was upregulated. HDAC1 overexpression or LSD1 silencing resulted in less VCN, mTOR inhibition and autophagy induction [52]. 

#### 4.3.3. Stimulating Autophagy in Vascular Calcification by Activating AMPK

The activation of AMPK with melatonin, metformin, intermedin_1-53_ or MOTS-c ameliorated VCN whereas the AMPK inhibitor compound C worsened VCN. Melatonin ameliorated VSMC calcification in vitro through the activation of AMPK [58]. In line with these findings, metformin ameliorated VSMC calcification in vitro through AMPK activation [59]. Additionally, both the peptide ghrelin and the protein intermedin_1-53_ (IM_1-53_) attenuated VCN through AMPK activation both, in vivo and in vitro [60,61]. Ghrelin induced both LC3-II and Beclin1 while IM_1-53_ upregulated Sirt1 expression. Also, MOTS-*c*, a mitochondrial derived peptide, ameliorated VCN in vivo by activating the AMPK pathway [62]. Moreover, compound C (an AMPK inhibitor) abrogated the protective effects of melatonin, metformin and ghrelin on VSMC calcification, as well as their effect on autophagy.

#### 4.3.4. Stimulation of Autophagy through Unspecified Mechanisms

A number of studies used experimental interventions that increase autophagy, while the exact molecular mechanisms where not investigated in detail. We have found reports of valproic acid, Polysaccharide from Fuzi (FPS), oestrogen, Nrf2, atorvastatin, miRNA-30b, ANCR that all increased autophagy and ameliorated VCN. Valproic acid, an autophagy inducer, attenuated VCN [51]. In contrast, intermittent administration of valproic acid did not ameliorate VCN [63]. Furthermore, FPS attenuated ox-LDL-induced VSMC calcification in vitro [64]. The authors stated that the protective effects of FPS on VCN are due to its ability to increase autophagy, characterized by an increased LC3-II/LC3-I ratio and decreased p62 expression. In vitro knockdown of Nrf2 impaired autophagy and increased VSMC calcification, while Nrf2 overexpression enhanced the LC3-II/LC3-I ratio and ameliorated VCN [65]. Furthermore, oestrogen inhibited VCN, increased Atg5, LC3-I and LC3-II, and its protective effects were counteracted by 3-MA. The authors therefore stated that the protective effects of oestrogen were autophagy dependent. Likewise, the knockdown of the oestrogen receptor alpha (ER_α_) inhibited autophagy and facilitated VCN [66]. Also, atorvastatin inhibited vascular calcification in vitro, while increasing Atg5, Beclin1 and the LC3-II/LC3-I ratio under calcifying conditions. More so, atorvastatin induced autophagy via the downregulation of the β-catenin pathway [67]. Furthermore, miRNA-30b attenuated VCN both in vitro and in vivo, whilst increasing both the LC3-II/LC3-I ratio and Beclin1 expression, as well as safeguarding the mitochondrial membrane potential. Therefore, the authors stated that miRNA30b protects against VCN by promoting autophagy through inhibition of Sox9 and mTOR [68]. Moreover, overexpression of anti-differentiation non-coding RNA (ANCR) induced autophagy, characterized by an increase in Atg5, LC3-I and LC3-II, and attenuated VCN both in vitro and in vivo [69]. Klotho knockout mice revealed age-related VCN, while beclin1 overexpression rescued Klotho knockout-induced VCN [70]. Next, iron citrate ameliorated VCN in vitro and enhanced autophagic flux [71]. Last, two studies investigated the effects of advanced glycation end-products (AGEs) on VCN and reported conflicting results. In one study, AGEs increased autophagic flux (demonstrated by an increased number of both autophagosomes and autolysosomes) and promoted VSMC calcification [72]. In another study, AGEs enhanced VCN, however, AGEs were shown to inhibit autophagy through AMPK inhibition and mTOR activation, resulting in decreased Beclin1 and LC3-II levels [73]. 

#### 4.3.5. Inhibiting Autophagy in Vascular Calcification

Several studies reported that inhibition of autophagy with 3-MA, Atg5/7 siRNA, lactate or lysophosphatidic acid (LPA) aggravates VCN. Dai et al. (2013a) showed that inhibiting autophagy with 3-MA or Atg5 knockdown enhances VCN in vitro. Other studies confirmed that 3-MA aggravates VSMC calcification [52,53,72]. Furthermore, multiple studies added 3-MA, in combination with an autophagy, inducing pharmacological intervention, thereby showing that 3-MA abolishes the beneficial effects of autophagy induction on VCN [59,61,64,66,67]. Additionally, CD137 activation led to JNK-mediated disrupted autophagic flux and aggravated VCN [74]. Furthermore, O-GlcNAc transferase (OGT) was upregulated in CKD-associated VCN, thereby, enhancing Yes-associated protein (YAP) glycosylation. The authors stated that elevated or in vitro overexpression of (glycosylated) YAP inhibits autophagic activity, facilitating VCN [75]. Next, excess amounts of saturated lysophosphatidic acid (LPA), a saturated fatty acid, were shown to inhibit the early steps of autophagosome formation, thereby, blocking autophagic flux. Moreover, saturated LPAs facilitated VCN while unsaturated LPAs counteracted these effects and restored autophagic activity [76]. Alongside general autophagy, mitophagy plays a role in VCN. Lactate aggravated VSMC calcification in vitro by inhibiting BCL2 and adenovirus E1B 19-kDa-interacting protein 3 (BNIP3)-mediated mitophagy, leading to mitochondrial dysfunction [77,78]. 

### 4.4. Lysosomal Function in Vascular Calcification

Impairment of lysosomal function has been shown to aggravate VCN. We have found a limited number of studies that interfered with lysosomal function by: (1) Deletion of *Mcoln1* or *Asah1* genes, (2) overexpression of the *Smpd1* gene, or (3) administration of 7-ketocholesterol (7-KC) aggravated VCN. However, lysosomal function was also shown to promote VCN by degrading the anti-calcifying protein granzyme B. Knocking out *Mcoln1* enhanced VCN both, in vivo and in vitro by impairing lysosomal trafficking, and consequentially enhancing secretion of pro-calcific small extracellular vesicles (sEVs) (Table 1) [79]. *Asah1* gene deletion also aggravated VCN and decreased TRPML1 activity, leading to impaired lysosomal trafficking and enhanced pro-calcific sEV secretion [80]. Additionally, sphingomyelin phosphodiesterase 1 (Smpd1) overexpression lead to increased VCN, impaired lysosomal trafficking, enhanced secretion of sEVs and increased arterial stiffness [81]. 7-KC was shown to aggravate VCN by inducing lysosomal dysfunction and autophagosome accumulation [82] Lastly, granzyme B was degraded in hypoxia-induced pulmonary calcification by chaperone-mediated autophagy, characterized by an increase in Heat Shock Protein Family A (Hsp70) Member 8 (HSPA8) and lysosome-associated membrane protein 2 (LAMP2A). Conversely, granzyme B overexpression alleviated VCN both in vivo and in vitro [83].

### 4.5. Autophagy in Valvular Calcification

Three studies used aortic valves from human patients, obtained after replacement surgery, to investigate the role of autophagy in AVCN (Table 2). In one study [84], *ULK1* and *MAP1LC3A* were decreased, while *BECN1*, *ATG3*, *ATG5*, *ATG7* and *ATG12* were increased in the aortic valves from patients with calcific aortic valve stenosis (CAVS). Additionally, *LAMP1* and cathepsin D (*CTSD*) were reduced, while an increase was observed in cathepsin B (*CTSB*), V (*CTSV*) and L (*CTSL*) in these patients. *TFEB* expression was unchanged. Further, the authors observed increased autophagic flux in VICs, isolated from the aortic valves from patients with CAVS. In contrast, another study [85] reported lower LC3-II expression in VICs from patients with CAVS. Additionally, 3-MA, bafilomycin A1 and ATG7 knockdown all increased BMP2 and alkaline phosphatase (ALP) levels, indicating enhanced VIC calcification. Moreover, ATG7 overexpression or rapamycin alleviated VIC calcification. A histological evaluation of aortic valves from patients with CAVS revealed an increased amount of ubiquitin labelling in calcified tissue. Therefore, the authors [86] stated that autophagic activity might be enhanced in these tissues. Finally, in the last study [87], LPS was combined with conditioned medium of bovine macrophages to enhance the effect of Pi-induced bovine VIC calcification. At high Pi concentrations, MAP1LC3 expression, as well as the total number of autophagic vacuoles decreased, whereas low-to-middle concentrations of Pi resulted in rough endoplasmic reticulum (RER)-dependent autophagic activity [87]. 

## 5. Discussion 

### 5.1. Inhibiting mTOR Protects against Vascular Calcification

Nine studies investigated the effect of AMPK or mTOR (or both) on vascular calcification (VCN) (Figure 3). One study (Zhan et al., 2014) reported increased mTOR activity in in vitro calcification and another study (Zhao et al., 2015a) found similar results in vivo. Moreover, mTOR overexpression has been shown to enhance VCN, while mTOR inhibition ameliorated VSMC calcification. Consequentially, rapamycin, adiponectin and bavachin have been shown to attenuate VCN due to their ability to inhibit mTOR activity [54,56,57]. Epigenetic regulation of mTOR activity has also been shown to modulate VCN. Under pro-calcifying conditions, HDAC1 and LSD1 expression were decreased, and increased, respectively [52]. LSD1 is implicated in various biological processes, such as cell proliferation, stem cell differentiation and enhancing mTORC1 activity [88]. Increasing HDAC1 activity or silencing LSD1 decreased mTOR activity and mitigated VCN [52]. These results indicate that mTOR plays an important role in the pathogenesis of VCN. A possible mechanism as to the reasons why mTOR contributes to the development of VCN, is via mTOR-mediated inhibition of *Klotho*. Notably, the Klotho protein, originally identified as an aging suppressor, possesses strong anti-calcification properties and prevents osteogenic trans differentiation of VSMCs [89]. One study [56] demonstrated that mTOR (over)expression inhibits Klotho expression whereas rapamycin alleviates this repression and ameliorates VSMC calcification. Interestingly, intermedin_1-53_ (IM_1-53_) also ameliorated VCN while the authors additionally observed a marked increase in *Klotho* expression after IM_1-53_ treatment. This increase in *Klotho* expression by IM_1-53_, is possibly the consequence of AMPK-dependent inhibition of mTOR [60]. Indeed, there are multiple lines of evidence supporting AMPK as an anti-VSMC calcification mechanism. Melatonin, metformin, ghrelin, intermedin_1-53_ and MOTS-c have all been demonstrated to ameliorate VCN in vitro and/or in vivo, accredited due to their ability to increase AMPK activity [58,59,60,61,62]. 

Collectively, these studies indicate that mTOR inhibition, either directly or resulting from AMPK activation, ameliorates VCN and prevents an osteochondrogenic phenotype switch of VSMCs. Whether mTOR modulates VCN through autophagy inhibition is less certain. mTOR has also been shown to enhance *Runx2* expression by modulating the oestrogen receptor alpha [90] in addition to suppressing the anti-calcification protein Klotho. Moreover, mTOR plays an important role in skeletal development, facilitating the progression of the preosteoblast stage to the mature osteoblast stage [91]. Besides the suppression of autophagy, mTOR has multiple other functions such as inhibition of cell growth and protein synthesis [92]. Hence, the protective effects of mTOR inhibition in VCN are generally accepted, these effects might partially be unrelated to autophagy. 

### 5.2. Autophagy Protects against Vascular Calcification

While, studies targeting mTOR in pro-calcifying conditions do not unambiguously prove the role of autophagy in VCN, inhibition of autophagy by 3-MA or by Atg5 silencing consistently increased VCN [51,52,53,72]. Also, other interventions inhibiting autophagy, such as CD137 activation, lead to aggravated VCN [74]. Furthermore, in pro-calcifying conditions, OGT is upregulated, enhancing YAP glycosylation. YAP glycosylation inhibits autophagy while YAP silencing ameliorates VCN [75]. Xu et al. proposed that YAP inhibits autophagy through activation of MST1 [75]. However, only the LC3-II/LC3-I ratio and p62 levels were measured, which is not sufficient to prove the effect of YAP on autophagic flux. Furthermore, another study demonstrated that saturated fatty acids (SFAs), which are increased in patients with CKD, inhibited autophagic flux drastically and facilitated VCN [76]. Collectively, these data demonstrate that autophagy is protective in VCN.

A few studies have investigated the role of mitophagy in VCN. Lactate was shown to inhibit BNIP3-mediated mitophagy, worsening VSMC calcification [77,78]. In contrast, metformin ameliorated VCN by inducing mitochondrial biogenesis through mitophagy [59]. Additionally, miRNA-30b ameliorated VSMC calcification and rescued mitochondrial membrane potential under calcifying conditions [68]. It is known that high Pi levels induce mitochondrial defects [93]. While, it can be expected that mitophagy plays a crucial role in mitigating aggravated oxidative stress, due to mitochondrial dysfunction [94], more research is warranted. 

Importantly, many studies used LC3-II/LC3-I ratios for evaluating autophagic flux. However, this ratio is rather unreliable and current guidelines on quantification of autophagic flux do not recommend the use of LC3-II/LC3-I ratios as a sole predictor of autophagic flux [95]. Furthermore, oestrogen, ANCR and atorvastatin have been demonstrated to enhance Atg5 expression and miRNA-30b has been shown to induce Beclin1 [66,67,68,69]. Only one study [71] determined autophagic flux (i.e., the turnover of LC3-II) by using an autophagy inhibitor, as is indicated in the most recent guidelines on the evaluation of autophagy [95]. Exemplary of the importance of using appropriate methodologies for evaluating autophagic flux are two studies investigating the effect of AGEs. While both studies confirmed that AGEs worsen VSMC calcification, conflicting conclusions with respect to autophagy induction were made. Liu et al. (2020b) stated that AGEs inhibit autophagy while Yang et al. (2019) reported increased autophagic flux. However, Liu et al. (2020b) only investigated autophagy indirectly, finding AMPK inhibition, activation of mTOR activity and decreased Beclin1 and LC3-II levels. In contrast, Yang et al. (2019) used mRFP-GFP-LC3 double labelling, which is a convenient and reliable method for monitoring autophagic flux, as it enables the simultaneous evaluation of autophagosomes (GFP-RFP) and autolysosomes (RFP) [95]. Moreover, the two research groups (Liu et al. and Yang et al.) used slightly different conditions, which could also explain their contrasting results. Nonetheless, these contradicting results emphasize the need for a more standardized approach for assessing autophagic activity. 

In general, inducing autophagy with pharmacological agents ameliorates VSMC calcification. While the selectivity of pharmacological agents used to induce autophagy is often limited, experimental studies with knockout of specific autophagy genes showed deterioration of VSMC calcification. Likewise, overexpression of Beclin1 protected against VCN. Therefore, there is a consensus that autophagy is protective in VCN and may be an attractive therapeutic target in VCN.

### 5.3. Intact Lysosomal Function Is Important for Controlling Vascular Calcification

While, hydroxyapatite itself damages cells and impairs lysosomes, multiple studies examined how impairing lysosomal function affects VCN (Figure 4) [55]. The TRPML1 channel mediates the fusion of lysosomes with autophagosomes. Notably, TRPML1 also regulates autophagy by regulating TFEB and by activating both calcium/calmodulin-dependent protein kinase kinase (CaMKKβ)-mediated signalling and ULK1 and hVPS34 [96]. Knocking-out *Mcoln1*, which is the gene that encodes TRPML1, resulted in increased VCN both in vitro and in vivo. Moreover, increased secretion of pro-calcific sEVs was reported in the absence of TRPML1 [79]. sEVs contain a variety of cargoes such as proteins and lipids that can chelate Pi and calcium. These calcium Pi-loaded vesicles tend to aggregate and form microcalcifications, acting as nucleation points and facilitating VCN [97,98]. sEVs are generated through different mechanisms, such as the fusion of multivesicular bodies (MVBs) with the plasma membrane or by budding, and consequent fission, of the plasma membrane [99]. Importantly, there is an inverse relation between the pathways that degrade MVBs and those that secrete sEVs. Indeed, it has been shown that inhibition of lysosomal degradation, enhances the release of sEVs [100]. Similarly, knocking out *Asah1*, the gene encoding for acid ceramidase (AC), enhanced sEV secretion and VCN. AC metabolizes ceramide into sphingosine (Sph) in lysosomes. Sph is crucial for proper functioning of the TRPML1 channel. A lack of AC impairs lysosome trafficking, leading to MVB accumulation and sEV release [101]. Last, the overexpression of lysosomal acid sphingomyelinase (ASM), encoded by *Smpd1*, aggravated calcification and also enhanced sEVs secretion [81]. An imbalance between ceramide production by ASM through sphingomyelin and ceramide consumption by AC, causes defective lysosomal acidification. Defective lysosomal acidification disrupts the function of TRPML1 and therefore impairs lysosomal trafficking [101,102]. Hence, sphingolipid metabolism is crucial for proper lysosome function and any defects in this pathway lead to aggravated VCN.

Granzyme B, a cytotoxic serum protease protein, was demonstrated to mitigate hypoxia-induced pulmonary arterial calcification (PAC) [83]. However, during hypoxia, granzyme B was degraded by chaperone-mediated autophagy (CMA). CMA is a proteolytic systems where intracellular proteins are degraded by lysosomes. CMA substrates are selectively targeted to the lysosome by the actions of chaperones and lysosomal membrane proteins such as LAMP1 [103]. Overexpression of granzyme B in PASMCs counteracted hypoxia-induced calcification [83]. This finding suggests that granzyme B plays a protective role in hypoxia induced PASMC calcification, whereas CMA aggravates calcification through granzyme B degradation.

Finally, 7-KC was shown to induce lysosomal dysfunction and oxidative stress, thereby aggravating VCN [82]. It is known that oxysterols such as 7-KC accumulate in the lysosomes, diminishing lysosomal proteolytic enzyme activity and resulting in inhibition of the fusion between lysosomes with autophagosomes or with endosomes [82]. Hence, both the autophagic machinery and the degradation of EVs are impaired, potentially attributing to the development of VCN. 

Taken together, there is compelling evidence that lysosomal function plays a critical role in mitigating VCN through inhibition of sEV secretion and reducing oxidative stress. Therefore, enhancing lysosomal function or lysosomal biogenesis could be a promising strategy for ameliorating VCN.

### 5.4. Protective Autophagy Is Upregulated in Aortic Valve Calcification

Aortic valve calcification (AVCN) is the major cause of valvular stenosis and shares many similarities to VCN, such as the transition of VICs to osteoblast-like cells and the formation of pro-calcific MVs [104]. However, little is known about modulating autophagy in valvular calcification. Three studies investigated aortic valves from patients with calcific aortic valve stenosis (CAVS) and all three studies found signs of enhanced autophagic flux. In one study [84], an upregulation of ATGs proteins was observed, as well as an increase in expression of lysosomal proteases. Also in vitro, using the LC3-II turnover assay, autophagic flux was enhanced in VICs [84]. In line with these findings, Somers et al. (2006) discovered more ubiquitin labelling in calcified tissue, possibly indicating enhanced autophagic activity. However, ubiquitin is not exclusive to the autophagy pathway, acting as a signal molecule for the ubiquitin-proteasome system as well [105]. Therefore, looking solely at ubiquitin does not provide sound evidence for claiming altered autophagic activity. Furthermore, inhibition of autophagy, either pharmacologically or genetically, aggravated VIC calcification [85]. Interestingly, the concentration of Pi seems to determine the type of autophagy during VIC calcification. Low-to-middle concentrations of Pi led to RER-dependent autophagy, whereas high doses drastically decreased the total number of autophagic vacuoles and MAP1LC3 [87]. All these findings indicated that, as in VCN, autophagy is upregulated and acts as a protective mechanism. However, more research is needed to gain more insight in how autophagy or lysosomal function is affected in valvular calcification and whether its upregulation or rescue is beneficial. 

### 5.5. Limitations

A notable weakness of this review is the heterogeneity between the studies. There are differences between in vivo and in vitro models. Furthermore, within in vitro studies there is considerable variation in the concentration of minerals, ranging from 1.4–5 mM for Pi (1.25–10 mM for β-GP) and 1.4–3 mM for calcium, which do not reflect physiological conditions. These experimental conditions should be taken into account when comparing the results, especially in the context of autophagy. Moreover, autophagy research is hampered by a lack of standardized methods for evaluating autophagy in different experimental settings. While guidelines for appropriate methods for investigating autophagy have been published (and are regularly updated), it became apparent that standardization of methods has not been achieved yet. Due to the large heterogeneity in models and methodologies, no meta-analysis was conducted. Finally, no quality assessment of the included studies was performed, given that the appropriate tools could not be found for this type of primary research. 

## 6. Concluding Remarks and Future Perspectives

Most studies (97%) reported a protective effect of autophagy and lysosomal function in VCN. Only one study [83] reported dissonant results. Pharmacological induction of autophagy was most often achieved through mTOR inhibition (directly or indirectly via AMPK activation) and consistently ameliorated VCN. However, the selectivity of current autophagy inducing pharmacological compounds is not absolute. Therefore, the development of novel, more selective autophagy inducers is crucial to realize the full therapeutic potential in VCN. Additionally, several recent studies indicate that lysosomes are crucial for mitigating VCN. Moreover, impaired lysosomal function aggravated VSMC calcification, but studies focusing on improving lysosomal function in VCN were not found. Future research is required to fully understand the role of lysosomal function in VCN and its possible synergy with autophagy. Finally, there is evidence that autophagy acts as a protective mechanism in AVCN as well. Yet, the data regarding this topic is sparse and more research is warranted. Besides, several methodological limitations still need to be overcome. Since autophagy is a dynamic and complex process, it is advised that future studies focus on measuring autophagic flux, with a view to standardisation.

## Figures and Tables

**Figure 1 ijms-21-08933-f001:**
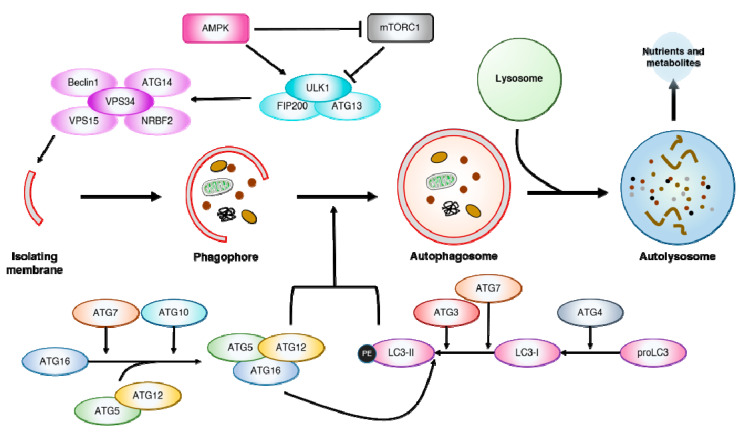
The autophagic process. The autophagic machinery starts with the formation of an isolation membrane, forming the double-membraned intermediate, the phagophore. The maturation of the phagophore into an autophagosome is dependent on the activity of two ubiquitin-like conjugation systems, the LC3-PE and Atg5-Atg12. In the first system, pro-microtubule-associated protein light chain 3 (proLC3) is cleaved by the cysteine protease Atg4 into LC3-I. LC3-I is then further processed into LC3-II and conjugated with phosphatidylethanolamine (PE) by Atg7 and Atg3, which are E1- and E2-homologous enzymes, respectively. LC3-II associates with the autophagic membranes with the aid of the Atg5-Atg12-Atg16 complex, facilitating phagophore maturation into an autophagosome. The Atg5-Atg12-Atg16 complex is formed by the action of Atg7 and Atg10, E1 and E2 enzymes respectively. This newly formed autophagosome then fuses with a lysosome forming an autolysosome. Here, lysosomal enzymes are responsible for the degradation of the autophagosome and its contents. Once degraded, the macromolecules (nutrients and metabolites) are secreted back into the cytosol. The shapes inside the phagophore and autophagosome are intra-cellular cargo such as (dysfunctional) mitochondria, (aggregated) proteins, etc. targeted for degradation; Atg = Autophagy related proteins.

**Figure 2 ijms-21-08933-f002:**
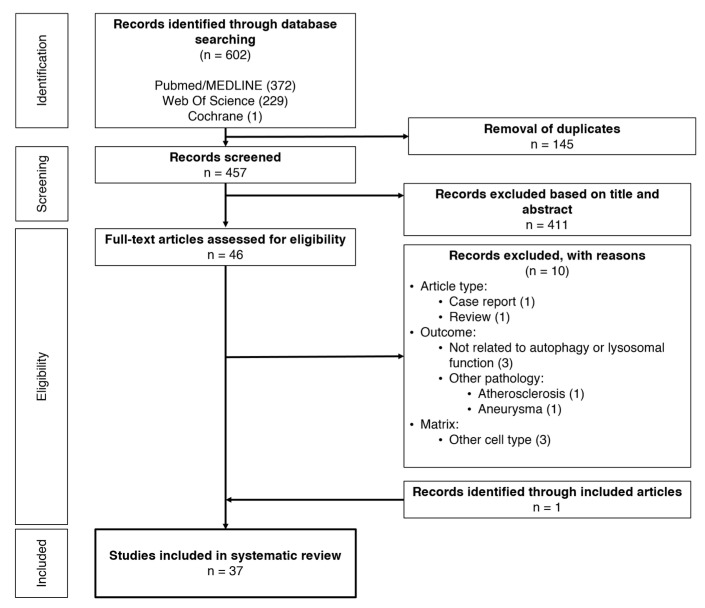
Review workflow.

**Figure 3 ijms-21-08933-f003:**
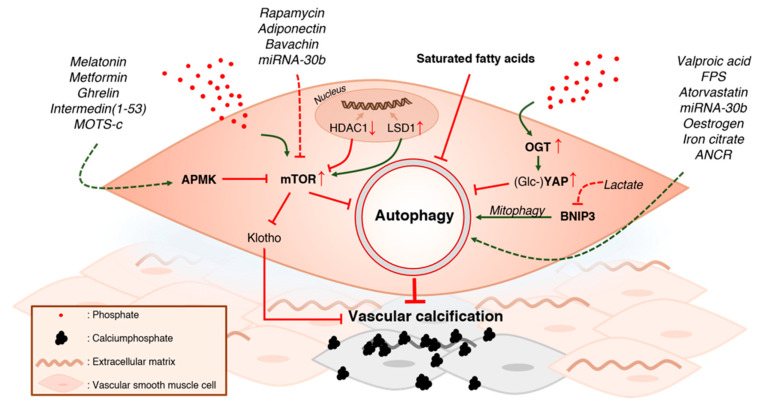
Modulating the autophagic machinery in vascular calcification. A high Pi environment promotes vascular smooth muscle cell (VSMC) calcification. This pro-calcific environment increases the activity of the mammalian target of rapamycin (mTOR), as well as O-GlcNAc transferase (OGT). The latter enhances YAP glycosylation. Additionally, epigenetic regulation of mTOR activity is also affected by the high Pi environment. Histone deacetylase 1 (HDAC1) is downregulated, while lysine-specific histone demethylase A1 (LSD1) is upregulated, resulting in enhanced mTOR activity. Both the increase in glycosylated (Glc) YAP and mTOR activity inhibit autophagy, facilitating VSMC calcification. Therefore, direct or AMPK-mediated inhibition of mTOR, through the use of different pharmacological agents, alleviates mTOR-dependent suppression of autophagy and ameliorates VSMC calcification. Saturated fatty acids, which are present in the pro-calcific environment, as seen in patients with chronic kidney disease, inhibit autophagy and are shown to facilitate VSMC calcification. Stimulation of autophagy with various pharmacological agents has been shown to attenuate VSMC calcification. Lastly, BNIP3-mediated mitophagy plays an important role in VSMC calcification, as inhibition of this pathway with lactate, promotes mineralization. AMPK = AMP-activated kinase, mTOR = mammalian target of rapamycin, HDAC1 = Histone deacetylase 1, LSD1 = Lysine-specific histone demethylase A1, OGT = O-GlcNAc transferase, Glc = Glycosylated, YAP = Yes-associated protein, BNIP3 = BCL2 and adenovirus E1B 19-kDa-interacting protein 3, ANCR = anti-differentiation non-coding RNA.

**Figure 4 ijms-21-08933-f004:**
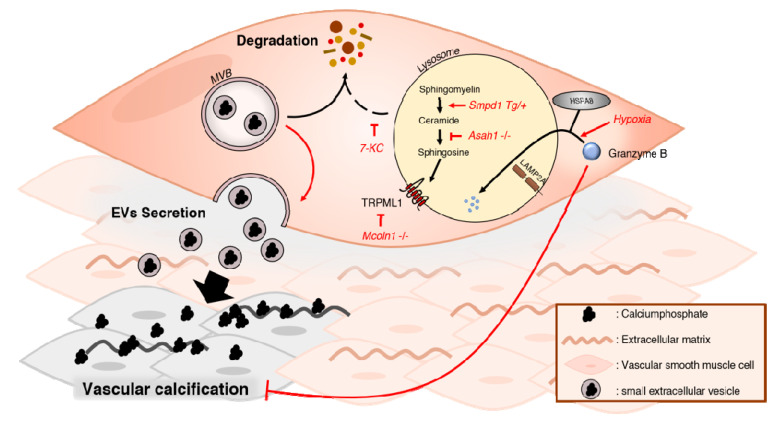
Modulating lysosomal function in vascular calcification. In a pro-calcific environment, calciumphosphate-loaded multivesicular bodies (MVBs), a type of late endosome, can either be degraded or secrete their contents to the extracellular matrix. In physiological conditions, there is a balance between degradation and secretion. Impairment of lysosomal function leads to decreased degradation of MVBs and enhanced secretion of pro-calcific extracellular vesicles (EVs). Transient receptor potential mucolipin 1 (TRPML1) is important for proper lysosome functioning, mediating the fusion of lysosomes with autophagosomes and endosomes. Knockout of the TRPML1 channel through the deletion of the mucolipin 1 gene, *Mcoln1*, impairs lysosome function, thereby enhancing the release of pro-calcific EVs and aggravating vascular calcification (VCN). Sphingosine is necessary for TRPML1 activity. Impairing the biosynthesis of sphingosine from ceramide, through the inhibition of the enzyme acid ceramidase, consequently impairs TRPML1 function, leading to enhanced sEV secretion and VCN. The balance between ceramide and sphingosine levels inside the lysosome dictates lysosomal function. Enhanced levels of ceramide through overexpression of the *Smpd1* gene, which encodes for acid sphingomyelinase, impairs lysosome function. Overexpression of *Smpd1* leads to enhanced secretion of EVs and VCN. Additionally, impairing lysosomal function with 7-ketocholesterol aggravates VCN. Finally, in a hypoxic environment, granzyme B is degraded via Heat shock protein family A (Hsp70) member 8 (HSPA8)-dependent chaperone-mediated autophagy. However, granzyme B has been shown to be protective against VCN. Therefore, chaperone-mediated autophagy worsens hypoxia-induced calcification. sEV = small Extracellular vesicle, MVB = Multivesicular bodies, HSPA8 = Heat shock protein family A (Hsp70) member 8.

**Table 1 ijms-21-08933-t001:** Studies on the modulation of autophagy and lysosomal function in vascular calcification.

Authors	Type of Intervention/Modulation	Methods/Matrix	Molecular Outcome	Study Conclusion
**Modulation of the autophagic machinery in vascular calcification**				
[51] (Dai et al., 2013b)	Non-pharmacological: Adenine diet-fed rats (CRF rats) Pharmacological: Autophagy inducers/inhibitors	In vivo Wistar rats—CRF rat model - Adenine (0.75%)-induced CRF In vitro Primary rat VSMCs and BASMCs Rat aortic ring calcification - CaCl_2_ (1.8 mM) + Pi (3 mM) induced calcification	CRF/high Pi: ↑Autophagic flux; ↑LC3-II, ↑VSMC calcification; ↑MV release 3-MA: ↑↑VSMC calcification; ↑↑MV release Atg5 siRNA (in vitro): ↑↑VSMC calcification Valproic acid: ↓VSMC calcification	Autophagy is upregulated in VCN and counteracts its progression, while autophagy inhibition has pro-calcification properties
[52] (Zhou et al., 2020)	Genetic: Histone deacetylase HDAC1 overexpression	In vivo Sprague-Dawley rats - Adenine (0.75%)-induced CRF + High Pi diet (1.3% phosphorus) In vitro Primary rat aortic VSMCs - CaCl_2_ (3 mM) + Pi (2 mM)-induced calcification	CRF rats/High Pi: ↑Calcification, ↓HDAC1, ↑LC3-II, ↓p62, ↑LSD1 HDAC1 overexpression: ↓Calcification, ↓Runx2, ↑α –SMA, ↑LC3-II, ↓p62, ↓LSD1 LSD1 silencing: ↓Calcification, ↓(p-p70S6K/p70S6K, p-rpS6/rpS6), ↑SESN2, ↑LC3-II, ↓p62 3-MA: ↑Calcification, ↑Runx2, ↓α –SMA Valproic acid: ↓Calcification, ↓Runx2, ↑α –SMA	HDAC1 overexpression attenuates VCN by inhibiting LSD1 via SESN2-dependent mTOR signalling
[53] (Frauscher et al., 2018)	Pharmacological: Rapamycin	In vivo DBA/2NCrl mice - High Pi diet (HPD; 20.2 g/kg phosphorus) induced calcification In vitro MOVAS cell line - CaCl_2_ (1.8 mM) + β-GP (1.25 – 2.5 mM) induced VCN	HPD/β-GP: ↑(*Trp53in*, *Igfbp3*, *Hmox1*, *Adrb2*, *Atg16l1*, LC3-II), ↑Vascular calcification Rapamycin: ↑↑LC3-II, ↓VSMC calcification, ↓*Runx2* 3-MA: ↑↑VSMC calcification	Uremic media calcification increases autophagy, which acts as a protective mechanism. Enhancing autophagy with rapamycin, attenuates VCN
[54] (Zhan et al., 2014)	Genetic and pharmacological: mTOR	In vitro Primary mouse aortic VSMCs - CaCl_2_ (1.8 mM) + β-GP (10 mM) induced calcification	β-GP: ↑mTOR mTOR-siRNA: ↓(ALP, OC), ↓Calcification Rapamycin: ↓(mTOR, p70S6k, p-mTOR^Ser2448^, p-p70S6k^Thr389^), ↓ALP, ↓Calcification Adiponectin: ↓mTOR, ↓ALP, ↓Calcification	mTOR is involved in the signal transduction of VCN, whilst inhibiting its activity attenuates VSMC mineralization and osteoblastic differentiation
[56] (Zhao et al., 2015b)	Pharmacological: Rapamycin Genetic: Klotho	In vivo Wistar rats - Adenine (0.75%)-induced CRF Klotho knockout mice (C57BL/6 – C3H/J background) In vitro Aortic rings (*Kl^−/−^* and WT mice) T/G HASMCs and BASMCs - CaCl_2_ (1.4 mM) + Pi (3 mM)induced calcification	High Pi: ↑(p-mTOR, p-S6K), ↓*Klotho* mTOR overexpression: ↑Calcification, ↓*Klotho* Kinase-dead mTOR: ↓Calcification Rapamycin: ↓Calcification, ↓(*Msx2*, *Cbfα-1*, *aggrecan*, *Sox9)*, ↑(M*gp*, *Opn*), ↑*Klotho* 3-MA: partially abrogates effects of rapamycin Klotho overexpression: ↓Calcification Klotho-siRNA: Abrogates effects of rapamycin	Rapamycin attenuates VCN by upregulating Klotho via mTOR inhibition
[57] (He et al., 2019b)	Pharmacological: Bavachin	In vitro Primary HASMCs - β-GP (10 mM) induced calcification	Bavachin: ↑(LC3-II, Beclin1), ↓p-mTOR, ↓(Runx2, BMP2, OPN, OPG), ↓(Wnt3A, β-catenin) Wortmannin: Abrogates the effect of bavachin Atg7 siRNA (in vitro): Abrogates the effect of bavachin	Bavachin suppresses HASMC calcification by acting on Atg7/mTOR-mediated autophagy signalling
[58] (Chen et al., 2020a)	Pharmacological: Melatonin	In vitro Primary rat VSMCs - CaCl_2_ (1.8 mM) + β-GP (10 mM) induced VCN	Melatonin: ↓(VCN, Runx2, ALP activity); ↑(LC3-II, Beclin1, p-AMPK, p-ULK1), ↓p-mTOR Compound C & MHY1485: Abrogated the effect of melatonin, ↑VCN	Melatonin attenuates VSMC calcification in vitro by acting on the AMPK/mTOR pathway
[59] (Ma et al., 2019b)	Pharmacological: Metformin	In vitro Primary rat aortic VSMCs - CaCl_2_ (1.8 mM) + β-GP (10 mM) induced calcification	Metformin: ↓VSMC calcification, ↓(Runx2, BMP2), ↑p-AMPK, ↓PDK4, ↑LC3-II/LC3-I, ↓p62, ↑(TFAM, NRF1, PGC-1a) Compound C: Abrogates protective effects of metformin Atg5 siRNA (in vitro): Abrogates protective effects of metformin	Metformin-mediated AMPK activation attenuates Pi-induced calcification and restores disrupted mitochondrial biogenesisMitophagy regulates metformin-induced mitochondrial biogenesis
[60] (Chen et al., 2020b)	Pharmacological: Intermedin_1-53_	In vivo Young and old SD rats - VitD (300,000 IU/kg)+ Nicotine (25 mg/kg)-induced VCN In vitro Primary rat VSMCs Mouse IMD^SMC−/−^ VSMCs Human VSMC CRL1999 cells - CaCl_2_ (2.5 mM) + β-GP (5 mM) induced VCN	IMD_1-53_: ↓VCN, ↓(Runx2, BMP2), ↑MGP, ↑(Sirt1, p-AMPK, p-AKT, p-PKA), ↑Klotho IMD^SMC−/−^:↑VCN, ↑Runx2, ↓Sirt1	IMD_1-53_ plays a protective role in VCN by upregulating Sirt1
[61] (Xu et al., 2017)	Pharmacological: Ghrelin	In vivo Sprague-Dawley rats - Vitamin D (300,000 IU/kg) + nicotine (25 mg/kg)- induced VCN In vitro Primary rat (SD) VSMCs - CaCl_2_ (1.8 mM) + β-GP (10 mM) induced calcification	Ghrelin: ↓VCN, ↓ALP activity, ↑(LC3-II, Beclin1), ↑p-AMPK 3-MA: Abrogates protective effect of Ghrelin Compound C: Abrogates the effect of Ghrelin on VCN and autophagy	Ghrelin attenuates VCN by inducing autophagy through AMPK activation
[62] (Wei et al., 2020)	Pharmacological: Mitochondrial-derived peptide MOTS-c	In vivo Sprague-Dawley rats - Vitamin D (300,000 IU/kg) + nicotine (5 mL/kg) induced VCN	MOTS-c: ↓VCN, ↓(Aortic calcium content, ALP activity), ↑p-AMPK, ↓(AT-1 receptor, ET-B receptor)	MOTS-c attenuates VCN by activating the AMPK pathway
[63] (Ciceri et al., 2015)	Non-pharmacological: intermittent suspension (IS) of Pi Pharmacological: Valproic acid (IS)	In vitro Primary rat VSMCs - CaCl_2_ (1.8 mM) + Pi (5 mM)-induced calcification	IS Pi: ↓VCN, ↓(Runx2), ↑(LC3-II), ↑Autophagic flux IS valproic acid: = VCN	Intermittent Pi increases autophagic flux and ameliorates VSMC calcification, intermittent valproic acid treatment, however, does not attenuate VCN
[64] (Liao et al., 2018a)	Pharmacological: Polysaccharide from Fuzi (FPS)	In vitro Primary human VSMCs, from femoral arteries - Ox-LDL induced VSMC calcification (1.8 mM CaCl_2_ + 10 mM β-GP)	Ox-LDL: ↑VSMC calcification, ↑*CBFA1*, ↓*SM22a*, ↓LC3-II/LC3-I, ↑p62 FPS: ↓VSMC calcification, ↓*CBFA1*, ↑*SM22a*, ↑LC3-II/LC3-I, ↓p62 3-MA: Abrogates protective effect of FPS on Ox-LDL induced calcification	FPS protects human VSMCs from Ox-LDL induced calcification, by activating autophagy
[65] (Yao et al., 2017b)	Genetic: Nrf2	In vitro Primary rat (SD) aortic VSMCs - CaCl_2_ (1.8 mM) + High Pi (1.4 – 2.5 mM)-induced calcification	Nrf2-siRNA: ↑Calcification, ↑(BMP2, Runx2), ↓Autophagosomes, ↓LC3-II/LC3-I Nrf2 overexpression: ↓Calcification, ↓(BMP2, Runx2), ↑Autophagosomes, ↑LC3-II/LC3-I	Activation of the Nrf2-ARE pathway alleviates hyperphosphatemia-induced calcification, possibly by inducing autophagy
[66] (Peng et al., 2017b)	Pharmacological: Oestrogen	In vivo C57BL/6 OVX mice - Vitamin D (500,000 IU/kg) induced calcification In vitro Primary mouse VSMCs - CaCl_2_ (1.8 mM) + β-GP (10 mM) induced calcification	Oestrogen: ↓Calcification, ↓(Runx2, ALP activity), ↑(Atg5, LC3-I, LC3-II) 3-MA: Counteracted effects of oestrogen ER_α_-antagonist/knockdown: ↑(Runx2, ALP activity), ↓(LC3-I, LC3-II)	Oestrogen-induced autophagy inhibits arterial calcification through the ER_α_ pathway
[67] (Liu et al., 2014)	Pharmacological: Atorvastatin	In vitro Primary rat (SD) VSMCs - CaCl_2_ (1.8 mM) + TGF-β1 (2 ng/mL)-induced calcification	Atorvastatin: ↓Calcification; ↓(ALP, BMP2, Osteocalcin), ↓Nuclear β-catenin expression, ↑Autophagy; ↑(Beclin1, Atg5, LC3-II/LC3-I ratio) Autophagy inhibitors (3-MA, Chloroquine, NH_4_Cl, bafilomycin A1): Suppresses effect of atorvastatin on autophagy and calcification β-catenin overexpression: Abrogates effect of atorvastatin β-catenin inhibitor JW74: ↑Effect of atorvastatin	Atorvastatin suppresses TGF-β1-induced VCN by inducing autophagy via downregulation of the β-catenin pathway
[68] (Xu et al., 2019b)	Pharmacological: miRNA-30b	In vivo Sprague-Dawley rats - (5/6 nephrectomy (Nx) + high Pi (1.2%) diet)-induced CKD In vitro VSMCs - (CaCl_2_ (1.8 mM) + β-GP (10 mM) induced calcification	miR-30b mimic: ↓VCN, ↓(SOX9, Msx2, Runx2) ↑(LC3-II/LC3-I, Beclin1), ↑Mitochondrial membrane potential (MMP) miR-30b inhibitor + Rapamycin: no mTOR inhibition	miRNA-30b protects against VCN by promoting MMP and autophagy, via targeted inhibition of SOX9 or negatively regulating the mTOR pathway
[69] (Zhang et al., 2020b)	Genetic: Anti-differentiation non-coding RNA (ANCR)	In vivo C57BL/6 mice - Calcitriol (500,000 IU/kg/day)induced arterial calcification In vitro Primary mouse aortic VSMCs - CaCl_2_ (1.4 mM) + β-GP (10 mM) induced calcification	ANCR overexpression: ↓Calcification, ↓(BMP2, Runx2), ↑(Atg5, LC3-II, LC3-I)	ANCR attenuates VCN and VSMC osteochondrogenic differentiation by activating autophagy
[70] (Shi et al., 2020)	Genetic: Klotho knockout Beclin1 overexpression	In vivo *Becn1F121A* knock-in mice (*129 S1/SVlmJ*) (*BK*) *αKlotho* homozygous knockout mice (*129 S1/SVlmJ*) (*kl/kl*) *BK/BK*; *kl/kl* mice - Age-induced calcification (10 weeks)	*αKlotho* homozygous knockout (*kl/kl*): ↑Calcification *Beclin1* homozygous overexpression: Attenuates calcification in α*Klotho* knockout mice	Beclin1 overexpression alleviates *kl/kl*-induced vascular calcification
[71] (Ciceri et al., 2019)	Non-pharmacological: Iron citrate	In vitro Primary rat aortic VSMCs - CaCl_2_ (1.8 mM) + Pi (5 mM)-induced calcification	Iron citrate: ↓Calcification, ↑Autophagosomes, ↑Autophagic flux, ↑LC3-Iiβ	Iron citrate blocks the progression of calcification by inducing autophagy
[72] (Yang et al., 2019)	Pharmacological: Advanced glycation end-products (AGEs)	In vitro Primary VSMCs - CaCl_2_ (1.8 mM) + β-GP (10 mM) induced calcification	AGEs: ↑HIF-1α, ↑PDK4, ↑LC3-II, ↓p62, ↑(Autophagosomes, autolysosomes),↑LC3-II & LAMP1 colocalization Dichloroacetic acid (PDK inhibitor) + AGEs: ↓LC3-II, ↑p62 Attenuates effect of AGEs Rapamycin: ↓Calcification, ↓Runx2 3-MA: ↑↑Calcification	AGEs induce autophagy through HIF-1α/PDK4 signalling, which has protective effects against AGEs-induced VCN
[73] (Liu et al., 2020b)	Pharmacological: Advanced glycation end-products (AGEs)	In vitro A7R5 cells Wild-type rat aortic segments	AGEs: ↑VCN, ↑(BMP2, RUNX2), ↓(BECN, LC3-II, p-AMPK↓), ↑p-mTOR	AGEs induce VSMC calcification by suppressing autophagy through action on the AMPK/mTOR signalling pathway
[74] (Chen et al., 2018)	Pharmacological: Agonist-CD137, SP600125	In vivo *ApoE^−/−^* mice - Western diet induced VCN In vitro Primary VSMCs from C57BL/6J mice - CaCl_2_ (1.8 mM) + β-GP (10 mM) induced VCN	Agonist-CD137: ↑AMC and osteogenic VSMC phenotype transition; ↑(Beclin1, p62, LC3B), autophagosome accumulation SP600125 (JNK-inhibition): Attenuated the effect of agonist-CD137	CD137 activation disrupts autophagic flux and accelerates calcification through the action of the JNK phosphorylation
[75] (Xu et al., 2020)	Genetic: O-GlcNAc transferase (OGT)	In vivo Sprague-Dawley rats - (5/6 nephrectomy (Nx) + high Pi (1.2%) diet)-induced CKD In vitro Rat VSMCs - CaCl_2_ (1.8 mM) + β-GP (10 mM) induced calcification	shOGT: ↓VCN OGT overexpression: ↑(YAP glycosylation, YAP protein expression & nuclear translocation), ↓YAP phosphorylation, ↓(Autophagosomes, LC3-II/LC3-I), ↑p62, ↑Calcification, ↑Runx2, ↓α –SMA Rapamycin or YAP silencing: ameliorates the effects of OGT overexpression YAP overexpression: ↓(Autophagosomes, LC3-II/LC3-I), ↑p62, ↑Calcification, ↑Runx2, ↓α -SMA	Upregulated OGT in high Pi diet-fed CKD rats, promotes glycosylation of YAP to inhibit autophagy, facilitating VCN
[76] (Shiozaki et al., 2020)	Non-pharmacological: (Un)saturated fatty acids (SFA/UFA)	In vivo *SMMHC*-CreER(T2); *Atg5*(lox/lox) mice SMC *Scd1/2* KO; *Gpat4* triple KO mice SMC-*Scd1/2* KO In vitro Human and mouse VSMCs, MOVAS-1 cell line GFP-LC3-RFP-LC3 G probe *Gpat4* siRNA - CaCl_2_ (1.8 mM) + Pi (2.6 mM) induced calcification	*Atg5* KO (in vivo/in vitro): ↑VCN, ↑p62, ↓LC3-II SFAs: ↓Autophagic flux (↑GFP/RFP), Accumulation of isolation membranes *Gpat4* KO: Attenuates autophagy inhibition by SFAs Unsaturated LPAs: ↓SFA-induced LC3-II accumulation, ↓Calcification SMC-*Scd1/2* KO: ↑Saturated LPAs, ↑↑Calcification, ↑(p62, Fam134b2) SMC-*Scd1/2* KO; *Gpat4* triple KO: Blocks SCD deficiency-induced VCN and autophagic flux inhibition	Excess saturated LPAs cause omegasome formation, which in turn produces and accumulates isolation membranes, blocks autophagic flux and causes VCNGpat4 converts SFAs to saturated LPAs, which makes Gpat4 an interesting target in VCN
[77] (Zhu et al., 2020b)	Non-pharmacological: Lactate	In vivo Wistar rats Vitamin D (300,000 IU/kg) + nicotin (25 mg/kg) induced VCN In vitro Rat aortic rings Primary rat (SD) aortic VSMCs - CaCl_2_ (1.4 mM) + β-GP (10 mM) induced calcification	Lactate: ↑Calcification; ↑(BMP2, Runx2), ↓α –SMA, ↓Autophagy/Mitophagy;↓(LC3-II, BNIP3), ↑p62, ↑Mitochondrial fission; ↑mito-Drp1, ↓OPA1, ↑NR4A1 *NR4A1* silencing/knockdown: Abrogates effects of Lactate, ↑Mitochondrial clearance/Mitophagy; ↑BNIP3, ↓(TOMM20, TOMM40), ↑Mitochondrial respiration, ↓mitochondrial swelling, ↑Autophagosome-mitochondria fusion	Lactate inhibits BNIP3-mediated mitophagy via the NR4A1/DNA-PKcs/p53 pathway, enhancing mitochondrial fission and therefore accelerating VCN
[78] (Zhu et al., 2019b)	Non-pharmacological: Lactate	In vitro Primary rat (SD) aortic VSMCs - CaCl_2_ (1.4 mM) + β-GP (10 mM) induced calcification	Lactate: ↑Calcification; ↑(BMP2, Runx2, ALP activity), ↓Mitochondrial function & biogenesis, ↓Autophagic flux; ↓Autolysosomes, ↓LC3-II, ↑p62, ↓Mitochondrial clearance/Mitophagy; ↑TOMM20, ↓BNIP3 BNIP3 overexpression: ↓Calcification; ↓(BMP2, Runx2)	Lactate accelerates VCN by facilitating mitochondrial dysfunction and inhibiting BNIP3-mediated mitophagy
**Modulation of lysosomal function in vascular calcification**				
[55] (Liu et al., 2020a)	Non-pharmacological: Hydroxyapatite (HAP)	In vitro A7R5 cells - Pi/HAP-induced calcification	Pi + HAP: ↑Calcification, ↑(Runx2, BMP2), ↓Lysosome integrity	Adhesion of HAP causes cell injury, which leads cell damage and decreased lysosomal integrity, therefore contributing to the development of VCN
[79] (Bhat et al., 2020c)	Genetic: Mcoln1^−/−^ Pharmacological: MLSA-1, Verapamil	In vivo Male *Mcoln1*^−/−^ mice (C57BL/6J) & controls - Vitamin D (500,000 IU/kg/bw/day) induced AMC In vitro Primary WT VSMCs - CaCl_2_ (1.8 mM) + Pi (3 mM) induced VSMC calcification, incubated +/- MLSA-1 or verapamil	Mcoln1^−/−^: ↑AMC and SMC osteogenic phenotype transition (in vivo), ↓Lysosomal trafficking/MVB colocalization, ↑sEV secretion Effect on lysosomal trafficking: Verapamil: ↓ MLSA-1: ↑	*Mcoln1*/TRPML1 deletion impairs normal lysosomal trafficking, leading to enhanced sEV secretion, contributing to the development of AMC
[80] (Bhat et al., 2020a)	Genetic: *Asah1^−/−^*	In vivo *Asah1^fl/fl^/SM^Cre^* (C57BL/6J) mice & controls - Vitamin D (500,000 IU/kg/bw/day) induced AMC In vitro Primary *Asah1^fl/fl^/SM^Cre^* CASMCs & controls - CaCl_2_ (1.8 mM) + Pi (3 mM) induced CASMC calcification	*Asah1*^−/−^: ↑AMC and SMC osteogenic phenotype transition, ↓Lysosomal trafficking/MVB colocalization, ↑sEV secretion, ↓in vitro TRPML1 channel activity	*Asah1* gene deletion impairs TRPML1 channel activation which causes impaired lysosomal trafficking, thereby enhancing AMC
[81] (Bhat et al., 2020b)	Genetic: sphingomyelin phosphodiesterase 1 (Smpd1) overexpression	In vivo *Smpd1 ^trg^/SM^cre^* mice (C57BL/6J) - Vitamin D (500,000 IU/kg/bw/day) induced AMC In vitro Primary mouse CASMCs - CaCl_2_ (1.8 mM) + Pi (3 mM) induced calcification	*Smpd1 ^trg^/SM^cre^* mice: ↑Calcification; ↑(OSP, Runx2), ↓SM22-α, ↓Lysosomal trafficking/MVB colocalization, ↑sEV secretion, ↑Arterial stiffness; disorganized elastic lamellae Amitriptyline: Abrogates effects of *Smpd1* overexpression	Lysosomal overexpression of *Smpd1* enhances the secretion of sEVs and facilitates osteogenic phenotype switch of SMCs, initiating arterial medial calcification
[82] (Sudo et al., 2015)	Pharmacological: 7-ketocholesterol (7-KC)	In vitro Human aortic SMCs - CaCl_2_ (1.8 mM) + Pi (3 mM) induced calcification	7-KC: ↑Calcium deposition/calcification, Autophagosome accumulation, ↑(p62, LC3-II), ↓Mature cathepsin B and D *Atg5* siRNA: = Calcium deposition ( vs. 7-KC) Lysosomal protease inhibitors: ↑↑Calcium deposition (vs. 7-KC)	7-KC induces oxidative stress through lysosomal dysfunction, aggravating HASMCs calcification
[83] (Mao et al., 2018)	Non-pharmacological and genetic: Granzyme B (GZMB)	In vivo Sprague-Dawley Rats C57BL/6 with GZMB overexpression (*SM22a-GZMB Tg*) - Hypoxia-induced pulmonary artery calcification (Hypoxic pulmonary arterial hypertension) In vitro Primary PASMCs - Hypoxia-induced SMC calcification	Hypoxia: ↑VCN, ↑(Runx2, BMP2, MSX2, SOX9), ↓GZMB, ↑Chaperone-mediated autophagy, ↑(HSPA8, LAMP2A) GZMB overexpression: ↓VCN	Chaperone-mediated autophagy degrades GZMB, which promotes pulmonary VCN

Abbreviations: 3-MA = 3-methyladenine; a-SMA = alpha-smooth muscle actin; AKT = protein kinase B; ALP = Alkaline phosphatase; AMC = Arterial medial calcification; AMPK = AMP-activated protein kinase; ARE = Antioxidant response element; AT-1 = Angiotensin II receptor type 1; BASMC = Bovine aortic smooth muscle cell; BECN = Beclin1; BMP2 = Bone morphogenetic protein 2; BNIP3 = BCL2 and adenovirus E1B 19-kDa-interacting protein 3; CBFA1 = Core-binding factor alpha 1; CKD = Chronic kidney disease; CRF = Chronic renal failure; Drp1 = Dynamin-1-like protein; ET-B = Endothelin B; Gpat4 = Glycerol-3-phosphate acyltransferase 4; HAP = Hydroxyapatite; HIF-1a = Hypoxia-inducible factor 1-alpha; HPD = High Pi diet; HSPA8 = Heat shock protein family A (Hsp70) member 8; JNK = c-Jun N-terminal kinases; LAMP2A = Lysosome associated membrane protein 2; LC3 = Light chain 3; LPA = Lysophosphatidic acid; LSD1 = Lysine-specific histone demethylase 1; MGP = Matrix Gla protein; Msx2 = Msh homeobox 2; mTOR = mammalian target of rapamycin; MV = Matrix vesicle; NR4A1 = Nerve growth factor IB; NRF1 = Nuclear respiratory factor 1; Nrf2 = Nuclear factor erythroid 2-related factor; OC = Osteocalcin; OGT = O-linked N-Acetylglucosamine transferase; OPA1 = Optic atrophy protein 1; OPG = Osteoprotegerin; OPN/OSP = Osteopontin; Ox-LDL = oxidized low density lipoprotein; p-AKT = phosphorylated AKT; p-AMPK = phosphorylated AMPK; p-mTOR = phosphorylated mTOR; p-PKA = phosphorylated protein kinase A; p-ULK1 = phosphorylated ULK1; PDK4 = Pyruvate dehydrogenase kinase 4; PGC-1a = peroxisome proliferator-activated receptor gamma coactivator 1-alpha; Pi = inorganic phosphate; PKA = Protein kinase A; RUNX2 = Runt-related transcription factor 2; Scd1/2 = Stearoyl-CoA desaturase-1/2; SD rats = Sprague Dawley rats; SESN2 = Sestrin 2; sEV = Small extracellular vesicle; SFA = Saturated fatty acid; Sirt1b = Sirtuin 1; SM22a = smooth muscle protein 22-alpha; SOX9 = SoxE group member SRY (sex determining region Y)- box 9; TFAM = Transcription factor A, mitochondrial; TOMM20/40 = Translocase of the outer mitochondrial membrane 20/40; TRPML1 = Mucolipin-1; UFA = Unsaturated fatty acid; ULK1 = Unc-51 Like Autophagy Activating Kinase 1; VCN = Vascular calcification; VitD = vitamin D; VSMC = Vascular smooth muscle cell; YAP = Yes-associated protein; β-GP = β-glycerophosphate.

**Table 2 ijms-21-08933-t002:** Studies on autophagy in valvular calcification.

Authors	Type of Intervention/Modulation	Methods/Matrix	Molecular Outcome	Study Conclusion
[84] (Carracedo et al., 2019)	Observational: Calcific Aortic Valve Stenosis (CAVS)	ex vivo Aortic valves from patients with CAVS In vitro Primary aortic valvular interstitial cells (VICs)	CAVS: ↓(ULK1, MAP1LC3A), ↑(BECN1, ATG3, ATG5, ATG7, ATG12), ↓(LAMP1, CTSD), ↑(CTSB, CTSV, CTSL), =TFEB VICs from CAVS patients: ↑Autophagic flux	Autophagy is upregulated in the aortic valves from CAVS patients as a pro-survival mechanism
[85] (Deng et al., 2017b)	Observational: Calcific Aortic Valve Stenosis (CAVS)	In vitro Primary human aortic VICs	VICs from CAVS patients: ↓LC3-II 3-MA, bafilomycin A1, ATG7 knockdown: ↑Calcification; ↑(BMP2, ALP) Rapamycin, ATG7 overexpression: ↓Calcification; ↓(BMP2, ALP)	Autophagy attenuates osteochondrogenic response in human AVICs
[86] (Somers et al., 2006)	Observational: Calcific Aortic Valve Stenosis (CAVS)	Ex vivo Aortic valves from patients with CAVS	CAVS: ↑Ubiquitin	Large amount of ubiquitinated cells are present in calcified aortic valve tissue, indicating possible higher autophagic activity
[87] (Bonetti et al., 2017)	Non-pharmacological: LPS	In vitro Bovine aortic VICs - Pi (+/− LPS and 20% bovine macrophage conditioned medium (CM))—induced calcification	LPS + CM: ↑Calcification, ↓Autophagic vacuoles, ↓RER hypertrophy, ↓MAP1LC3	High Pi concentrations lead to pro-calcific cell death, while low/middle Pi activate RER-dependent autophagic activity

Abbreviations: (MAP1) LC3 = (Microtubule-associated proteins 1A/1B) light chain 3B; ALP = Alkaline phosphatase; ATG = Autophagy related; BECN1 = Beclin-1; BMP2 = Bone morphogenetic protein 2; CTS = cathepsin; LAMP1 = Lysosome associated membrane protein 1; LPS = Lipopolysaccharide; RER = Rough endoplasmic reticulum; TFEB = Transcription factor EB; ULK1 = Unc-51 like autophagy activating kinase 1.

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
