# Peer review of "The Protective Effects of the Autophagic and Lysosomal Machinery in Vascular and Valvular Calcification: A Systematic Review"

_ijms, 2020, doi:10.3390/ijms21238933_

Round 1
Reviewer 1 Report
The article submitted by Neuten et. al is a well-written systematic review detailing the roles of autophagic and lysosomal machinery in vascular and valvular calcification.
The search strategy as well as the inclusion and exclusion criteria are clearly described. The 37 papers that are included in this review are organized in two tables that makes it easy to quickly extract all the most relevant information from the reviewed publications. Besides the tables, two figures help to understand the modulations of the autophagic and lysosomal machinery in vascular calcification. Although the publication date was not an exclusion criterion, most of the 37 reviewed papers (92%) were published in the last 5 years and 58% of the papers published between 2019 and 2020. This shows that this topic is relatively new and active, and that the review is up to date. The structure and flow of this review article are well-presented, and the citations are appropriate as well. I have no major concerns.
Author Response
Reviewer 1 had no comments and we would like to thank him/her for his/her time to evaluate our manuscript.
Reviewer 2 Report
Thank you for your interesting review presented to this journal. The review provides a comprehensive knowledge regarding the modulation of the autophagy in both vascular calcification and aortic valve calcification. The review has been well designed, the information is well organized and provides enough information to better understand the phenomenon of autophagy in calcification situations.
There are a few minor concerns These are contributions that could improve the revision.
I do not understand the criterion that a new reference has been provided. The review inclusion criteria do not indicate that the references of the selected articles have also been considered. It is not clear to the reader what the criterion of including this reference is.
The Table 1 should be reorganized in three sections, one with in vitro studies in cells, the second about studies in tissues and the third all in vivo studies. This aspect is left to the authors' discretion.
I miss the indication of the concentrations of calcium and phosphate used in in vitro studies, since these are often very far from physiological concentrations. Could it be that these non-physiological concentrations affect autophagy?
Author Response
First, we would like to thank the reviewer for the valuable feedback and his/her time for assessing our manuscript. We have answered his/her considerations point-by-point.
- Comment 1: I do not understand the criterion that a new reference has been provided. The review inclusion criteria do not indicate that the references of the selected articles have also been considered. It is not clear to the reader what the criterion of including this reference is.
Response: In systematic reviews it is common practice to include cited references (see PRISMA guidelines: https://doi.org/10.1136/bmj.b2700 ). To clarify this comment, we have added the following text in the “methods” section (page 4, line 143): “The reference lists of the included articles were checked to identify additional relevant studies”.
- Comment 2: The Table 1 should be reorganized in three sections, one with in vitro studies in cells, the second about studies in tissues and the third all in vivo This aspect is left to the authors' discretion.
Response: While this is a good suggestion, most of the included studies do not fit in a single category. Separating in vivo from in vitro studies would cause a lot of repetition. Moreover, a lot of studies make their conclusions based on the combined results from their in vivo and in vitro studies. With the suggested changes, this combined interpretation would be lost. Therefore, we opted for a lay-out where all the information of one study is displayed together.
- Comment 3: I miss the indication of the concentrations of calcium and phosphate used in in vitro studies, since these are often very far from physiological concentrations. Could it be that these non-physiological concentrations affect autophagy?
Response: We followed the suggestion of the reviewer and added the concentrations of the minerals (both phosphate and calcium) in table 1. Additionally, we added the following text to the limitations section (page 23, line 471): “There are differences between in vivo and in vitro models. Furthermore, within in vitro studies there is considerable variation in the concentration of minerals, ranging from 1.4 – 5 mM for Pi (1.25 – 10 mM for β-GP) and 1.4 – 3 mM for calcium, which do not reflect physiological conditions. These experimental conditions should be taken into account when comparing results, especially in the context of autophagy.”